# Bovine Pluripotent Stem Cells: Current Status and Prospects

**DOI:** 10.3390/ijms25042120

**Published:** 2024-02-09

**Authors:** Lanxin Chen, Bo Tang, Guanghong Xie, Rui Yang, Boyang Zhang, Yueqi Wang, Yan Zhang, Daozhen Jiang, Xueming Zhang

**Affiliations:** State Key Laboratory for Zoonotic Diseases, College of Veterinary Medicine, Jilin University, Changchun 130062, China; chenlx21@mails.jlu.edu.cn (L.C.); tang_bo@jlu.edu.cn (B.T.); xiegh@jlu.edu.cn (G.X.); ruiyang22@mails.jlu.edu.cn (R.Y.); zby23@mails.jlu.edu.cn (B.Z.); yueqiw22@mails.jlu.edu.cn (Y.W.); z_yan22@mails.jlu.edu.cn (Y.Z.); jiangdz23@mails.jlu.edu.cn (D.J.)

**Keywords:** cattle, embryonic stem cells, potential stem cells, induced pluripotent stem cells, reprogramming

## Abstract

Pluripotent stem cells (PSCs) can differentiate into three germ layers and diverse autologous cell lines. Since cattle are the most commonly used large domesticated animals, an important food source, and bioreactors, great efforts have been made to establish bovine PSCs (bPSCs). bPSCs have great potential in bovine breeding and reproduction, modeling in vitro differentiation, imitating cancer development, and modeling diseases. Currently, bPSCs mainly include bovine embryonic stem cells (bESCs), bovine induced pluripotent stem cells (biPSCs), and bovine expanded potential stem cells (bEPSCs). Establishing stable bPSCs in vitro is a critical scientific challenge, and researchers have made numerous efforts to this end. In this review, the category of PSC pluripotency; the establishment of bESCs, biPSCs, and bEPSCs and its challenges; and the application outlook of bPSCs are discussed, aiming to provide references for future research.

## 1. Introduction

“Pluripotency” refers to the potential of a cell to differentiate into any cell type in the body [1]. Initially, scientists observed the pluripotency of pluripotent stem cells (PSCs) during human embryonic development. PSCs include embryonic stem cells (ESCs) from preimplantation embryos, induced pluripotent stem cells (iPSCs) obtained through somatic cell reprogramming, and expanded potential stem cells (EPSCs), which all have the basic advantage of self-renewal and differentiation [2]. Self-renewal means that PSCs can generate new cells unlimitedly, and these new cells have the same bioproperties as their ancestors. The differentiation capacity means that PSCs can differentiate into three germ layers and multiple cell types [3]. These properties make them uniquely attractive for engineering genomes. PSCs can be used for genome modification, screening of gene-edited cells, generating embryonic chimeras and passing the engineered genome to the next generation, contributing to germline optimization, and improving reprogramming efficiency by somatic cell nuclear transfer (SCNT) [4].

ESCs are derived from the inner cell mass (ICM) of the embryo at an early stage. They manifest pluripotency by forming three germ layers, which can develop into organ-like structures in vitro. However, cells in the ICM are unable to self-renew due to a globally hypomethylated genome, which makes the embryonic pluripotency transient. In contrast, artificial cultures allow embryonic pluripotency to be maintained in vitro. Pluripotent cells cultured in a stable system can self-renew indefinitely and retain the ability of multilineage differentiation. The methods to maintain pluripotency include in vitro culture of embryonic germ cells, SCNT, and cell reprogramming (restoring pluripotency of differentiated cells) [5,6]. These in vitro-cultured pluripotent cells exhibit molecular and functional properties similar to the embryonic ectoderm. As the first established PSCs, ESCs were initially developed from mouse ICM and have been subsequently explored in other species, including humans. From an agricultural perspective, the establishment of ESCs in domestic animals helps generate transgenic livestock. In addition, ESCs are a valuable resource in biotechnology and biomedicine as experimental models for studying incurable or inherited diseases and developing therapies. Although ESCs provide great potential for multiple fields and numerous efforts have been made, they are less successful in domestic animals, especially in large livestock like cattle [7]. 

The development of iPSCs opens up new possibilities for practical applications of PSCs. Mature cells and the enucleated oocytes can be recombined into zygotes after nuclear transplantation, demonstrating the potential of somatic cells to be reprogrammed into PSCs. By ectopic expression of a few key transcription factors, somatic cells can be reprogrammed to iPSCs similar to ESCs. The pluripotency, proliferative capacity, viability, and genomic demethylation of iPSCs are activated and maintained. Thus, iPSCs are able to differentiate into multicellular lineages in vitro and form teratomas and chimeras in vivo. Additionally, iPSCs have the advantages of ESCs but raise no ethical concerns. These features expand the use of stem cells in regenerative medicine [8]. iPSCs have been extensively studied and characterized in humans and mice. Several iPSC-like cells have also been established from cattle, but reproducible results still need to be improved due to interspecies differences. The underlying mechanisms of pluripotency activation usually follow a similar pattern in mammals; thus, the recent advances in iPSC technology could be used to facilitate the establishment of bovine iPSCs (biPSCs). The chimeric livestock generated with iPSCs might be employed as biomedical models and change the status of agricultural science and technology [9].

## 2. Category of PSC Pluripotency

Mouse preimplantation embryos have been studied more intensively than those of other mammals (e.g., cattle). Generally, preimplantation development is the only developmental stage that can be studied in vitro in most mammals. In a mouse preimplantation embryo, the totipotent zygote gives rise to the first three cell lineages of the embryo, the trophectoderm, epiblast, and primitive endoderm. From mouse to human, most mammals follow a general sequence of events consisting of fertilization followed by early development without embryonic transcription; cleavage divisions in the absence of growth; compaction to form a morula; and finally, generation of a blastocyst. During this mitotic cleavage process, there are increasingly distinct differences between the blastomeres (the new cells in early embryos). Three rounds of cleavage are followed by the formation of an eight-cell embryo which undergoes polarization and densification at this stage. Subsequently, prominent cell differentiation occurs during the morula stage, followed by the internalization of some blastomeres to form the pluripotent ICM. In contrast, other blastomeres remain outside the embryo to form the trophectoderm [10]. The emergence of these two types of cells represents the first genealogical differentiation during embryo development. When the embryo develops into the blastocyst, the ICM differentiates into the epiblast and primitive endoderm. The developmental program of the morula is not predetermined but consists of varying division patterns and flexibility of cell fate that can compensate for cell loss [11]. Cells derived from the early ectodermal ICM transition from the naïve state to the primed state [12]. Therefore, PSCs can be categorized into three types: naïve, formative, and primed. The naïve PSCs correspond to the blastocyst ICM stage, and the primed PSCs correspond to the post-implantation ectoderm stage [13]. The naïve PSCs are capable of forming somatic chimeras and germline chimeras. The primed PSCs have a biased differentiation potential and weak chimerism under conventional conditions. The formative PSCs are in-between, chimerism-capable, but less efficient than naïve PSCs, and can make a direct response to germ cell induction [14]. The PSCs in different states differ in cell morphology, gene expression, epigenetic features, and developmental potential [15]. Compared with the primed PSCs and formative PSCs, the naïve PSCs have more advantages. They have a higher proliferative capacity and lower requirements for culture environments. They also have high plating efficiency when dispersed into single cells. Therefore, they are the most desirable PSCs. It has been demonstrated that naïve-like PSCs obtained by the conversion of primed PSCs have similar advantages to naïve PSCs and can be used as an important model for the study of differentiation pathways. Naïve and primed PSCs can also be converted into each other and passaged stably under appropriate culture conditions (Figure 1).

## 3. Establishment of Bovine ESCs (bESCs)

The program to establish bESCs is mainly based on mouse and human programs. Mouse ESCs (mESCs) were obtained from mouse embryos in 1981 [16], and human ESCs (hESCs) derived from human blastocysts were established in 1998 [17]. Researchers found that mESCs and hESCs differ in culture conditions and characteristics. mESCs depend on leukemia inhibitory factor (LIF) and bone morphogenetic protein 4 (BMP4) signaling and present as dome-shaped colonies, which can proliferate indefinitely. In contrast, hESCs depend on fibroblast growth factor 2 (FGF2)/activin signaling, have a flattened colony morphology, and are incapable of single-cell passaging. Mouse epiblast stem cells (mEpiSCs) are derived from post-implantation embryos and have similar characteristics to hESCs [18]. Researchers have classified mESCs as a naïve state, hESCs and mEpiSCs as a primed state. Naïve ESCs express pluripotency genes, such as *OCT4*, *SOX2*, and *NANOG*. These cells are able to differentiate into three germ layers rapidly after being injected into the host blastocysts. In naïve ESCs, the X chromosome is activated and this activation is an epigenetic hallmark of PSCs. Using glycogen synthase kinase-3beta (GSK-3β) inhibitor CHIR99021 and mitogen-activated protein kinase (ERK) inhibitor PD0325901, this so-called “2i” can maintain mESCs in the naïve state. Primed ESCs are at a more advanced developmental stage than naïve ESCs and also express pluripotency genes, which have similarities but not identical gene expression profiles [19]. 

According to the above-mentioned description, bESC-like cells were established by various methods such as in vitro fertilization (IVF) and SCNT. Researchers have attempted to apply culture systems of mESCs and hESCs directly to bESC-like cells. In 1992, SAITO first obtained bESC-like cells using the mESC system, the ICM isolated from in vivo-derived embryos, feeder layer cells, LIF, and other growth factors [20]. In 2005, WANG also obtained bESC-like cells by using the culture system of hESCs supplemented with bFGF [21]. However, long-term studies have found that it is not feasible to establish bESCs directly using mouse and human culture systems, and most of the reported bESC-like cell lines were able to form embryoid bodies in vitro and teratomas in vivo, but the chimera formation was not reported [22]. In addition, these bESC-like cell lines have poor derivation efficiency and limited proliferative capacity and no longer express pluripotency markers after prolonged passages. This is probably due to hoofed animals, rodents, and primates sharing only major pluripotent epigenetic traits.

Microarray analysis of the above-mentioned bESC-like cells revealed that bovine pluripotency-associated genes are closely related to the TGF-β, WNT, and LIF signaling pathways and also might be associated with BMP signaling. LIF inhibits the differentiation activity of mESCs, maintains them in a totipotent state, and stimulates their self-renewal. STAT3 is the essential signaling transducer after LIF activation and mediates most of the cellular effects [23]. Unlike in mice, signaling between LIF and STAT3 may be disconnected in cattle, which is possibly the reason why the true bESCs could not be generated in the previous culture system using LIF. This suggests that STAT3 may be the key molecule in establishing bESCs [24]. Inactivation of the WNT pathway is essential for the development of preimplantation and early post-implantation embryos and stabilizes the cytoplasmic β-linker proteins by inhibiting GSK-3β. The addition of classical WNT inhibitors may be critical for the derivation and proliferation of bESCs. bFGF is beneficial to maintain bESCs. The LIF and ERK inhibitor PD98059, known to promote pluripotency of mESCs, is ineffective in the maintenance of bESCs. The GSK3β inhibitor CHIR99021 and ERK inhibitor PD0325901 effectively promote the expressions of stem cell markers in bESCs [25]. However, the complete signaling pathway for the reprogramming and maintenance of bESCs has still not been reported. In 2018, Bogliotti et al. developed a culture system supplemented with inhibitors of the FGF2 and WNT signaling pathways and obtained stable primed bESCs. These primed bESCs are able to be passaged for an extended period and maintain their normal morphology, karyotype, transcriptome, and epigenetic features after 70 generations. The authors also derived teratomas but not chimerism [26]. Subsequently, Soto et al. conducted an in-depth study of this culture system, optimizing it for a more straightforward one that does not require a feeder layer [27]. Their study made bESC establishment more standardized and broadened the potential application of these special cells (Figure 2).

Compared with rodents and primates, it is still complicated and hard to establish stable bESCs. Continued optimization of the culture system is still needed, so that it can provide a gene network that stably maintains bESC pluripotency. Currently, the establishment of bESCs still depends on the systems of hESCs and mESCs, and new options need to be explored as well. Once established, bESCs might be expected to facilitate genome editing research, accelerate the process of molecular breeding for economic traits, and provide a new platform for the study of preimplantation embryonic development in cattle. Additionally, large farm animals are physiologically and morphologically similar to humans and can be models for clinical stem cell therapy and human diseases.

## 4. Establishment of Bovine iPSCs (biPSCs)

In 2006–2007, Yamanaka’s team reprogrammed mouse adult fibroblasts into iPSCs by overexpressing four transcription factors (*POU5F1*, *SOX2*, *KLF4*, and *c-MYC*) [32]. Subsequently, human iPSCs were also obtained from fibroblasts using the same method [33]. Since then, scientists have attempted to apply the method to livestock such as pigs, cattle, sheep, goats, horses, and dogs to obtain iPSCs [34]. The biPSC-like cells were first established in 2011 from bovine embryonic fibroblasts [35]. Currently, the method of reprogramming bovine somatic cells to biPSCs is developed continuously. The biPSC-like cells that can be stably passaged for 50 generations were established by Talluri et al. in 2015 [36]. In 2017, single-cell-derived biPSC-like cells that can be stably passaged for more than 50 generations were obtained, which can express essential pluripotency genes and form embryoid bodies and teratomas [37]. We also derived biPSC-like cells from bovine immature Sertoli cells recently [38,39]. Generally, the main factors affecting the quality of biPSCs include vectors, transcription factor combinations, culture systems, and somatic cell sources.

### 4.1. Vectors

The selection of appropriate vectors is critical in determining the success of reprogramming. The use of retroviral and lentiviral vectors has been widely reported. The advantages of using viral vectors include high efficiency and simple as well as easy operation. However, the use of viral vectors causes the reprogramming genes to be continuously expressed in the cells, which possibly hinders the study of further differentiation of iPSCs. Meanwhile, the safety of the iPSC-like cells is also worth considering, because their introduction into animals is likely to induce tumors/cancers [40]. To compensate for these drawbacks, scientists have used Sleeping Beauty (SB) and piggyBac (PB) transposon systems, which allow reprogrammed genes to be delivered and expressed transiently within the cell and minimize the possibility of genomic integration [36]. There are also reports of using non-viral delivery and protein transduction [41]. Each method has advantages and disadvantages, and overall their reprogramming efficiency is low.

### 4.2. Transcription Factors

Different combinations of transcription factors may improve the efficiency and quality of biPSC generation. The combinations of *OCT4*, *KLF4*, *SOX2*, and *C-MYC* (OSKM) [42]; *OCT4*, *KLF4*, *SOX2*, *C-MYC*, and *NANOG* (OSKMN) [43]; and *OCT4*, *KLF4*, *SOX2*, *C-MYC*, *NANOG*, and *LIN28A* (OSKMNL) have been commonly used. Most reports have used a protocol with at least four transcription factors, but there are also reports that biPSC-like cells can be generated from testicular germ cells using only *OCT4* [44]. The efficiency of reprogramming bovine fibroblasts into biPSC-like cells has been significantly improved by adding SV40 large T antigen [45]. Additionally, biPSC-like cells were obtained from bovine mesenchymal stem cells using a combination of overexpression of lysine-specific demethylase 4A (KDM4A) and other reprogramming factors [46]. This means that new reprogramming factors related to pluripotency can improve reprogramming efficiency. Human, mouse, and bovine transcription factors have been frequently used for reprogramming [47,48]. Homology analysis of mouse, human, and bovine transcription factors and proteins revealed that bovine and human homologies are close to each other. Reprogramming using bovine-derived transcription factors might improve the efficiency of bovine somatic cell reprogramming. Molecularly, reprogramming remodels somatic transcriptomic and chromatin programs to the state of ESCs. This includes processes such as gene silencing and X chromosome activation, demethylation of *OCT4* and *NANOG* promoter regions, and genome-wide resetting of histone H3 lysine 4 and 27 trimethylation [49,50,51]. A key question is how these transcription factors function to make such changes. Researchers are constantly exploring unknown reprogramming factors and trying to understand the mechanisms by which they work. Recently, it has been found that remote chromatin interactions play an important role in regulating cell fate decisions. CCCTC-binding factor (CTCF) is a key regulator in the reprogramming process, with a two-stage action. The first stage is the up-regulation of CTCF expression, which acts as a chromatin insulator to repress genes in somatic cells, thereby reducing the interaction between enhancer–promoter and somatic genes. The second stage is that CTCF helps maintain chromatin accessibility and activate pluripotency genes [52]. However, it is unclear how the different reprogramming stages are regulated to accomplish the reprogramming. Gene expression can be regulated at every stage of transcription. Transcriptional regulation acts as an on/off switch. Post-transcriptional mechanisms act as a rheostat to improve the output of gene expression, which plays a critical role in pluripotency through RNA-binding proteins, RNA-processing mechanisms, and RNA regulatory molecules. The study of post-transcriptional mechanisms can help obtain higher-quality biPSCs [53].

### 4.3. Chemical Small Molecules

Traditional reprogramming techniques drive direct cell fate shifts by overexpressing endogenous transcription factors, making the induction process difficult to control. Chemical reprogramming, on the other hand, utilizes exogenous chemical small molecules to mimic external signaling stimuli to drive cell fate transformation in a staged manner. This approach is highly controllable and promises to regulate cell fate accurately and reverse cell identity as well as its functional status, making the reversed development possible. The chemical reprogramming approach has made significant progress in both mice and humans [54,55]. This technique may also become a new direction in the development of bovine somatic cell reprogramming.

### 4.4. Culture System

The culture system is the key to obtaining high-quality biPSCs, by ensuring their stability and maintaining their pluripotency, so it is essential to explore the combination of various growth factors and their concentrations. Scientists have made great efforts in the composition of the culture system. Naïve and primed biPSCs have different gene expression profiles and maintain self-renewal through different signaling pathways [56]. bFGFs trigger protein kinase C (PKC) signaling by self-phosphorylation and activation of FGF receptors, thus regulating cell self-renewal, metabolism, survival, proliferation, and differentiation [57]. On the other hand, LIF inhibits the differentiation activity of iPSCs and promotes cellular self-renewal, activating pluripotency through three pathways: STAT3, PI(3) kinase, and MAPK. In addition, inhibitors PD0325901 and CHIR99021 inhibit MAPK and GSK-3β signaling, thereby suppressing the expression of FGF receptors and DNA methyltransferases to maintain naïve pluripotency [58]. Reversal of FGF2 action is critical in controlling the interconversion of primed and naïve states and vice versa. Combining LIF with FGF under inhibitor-free conditions to block GSK-3β and MEK signaling resulted in primed biPSCs, and naïve biPSCs were obtained when LIF was used in combination with CHIR99021 and PD0325901(2i) medium, or with BMP4 [59]. Using an autologous feeder layer may circumvent some of the challenges in the composition of the medium [60,61]. Our work showed that using a bovine embryonic fibroblast feeder layer enhances the reprogramming of bovine somatic cells, and thus is more conducive to the long-term maintenance of the colony morphology and pluripotency than using an MEF feeder layer. This method also improves the efficiency of the conversion of primed biPSC-like cells to naïve biPSC-like cells [39]. Successful conversion systems of primed iPSC-like cells to naïve iPSC-like cells in other species including humans provide more lessons for cattle [62,63]. In addition, Bessi et al. found that a hypoxic environment was detrimental to maintaining pluripotency in biPSC-like cells by comparing oxygen levels during and after reprogramming [64]. Overall, a more specific and effective culture system needs to be developed for biPSCs.

### 4.5. Source Cells

The difficulty to induce somatic cells from different tissues/organs entering the “critical state” varies greatly, since source cells have different gene modifications and expression patterns. Therefore, the somatic origin of iPSCs affects the efficiency of reprogramming. Moreover, the differentiation status of the cells is a central factor in reprogramming efficiency, and progenitor cells are more likely to be reprogrammed than differentiated terminal cells [65]. biPSC-like cells have been derived from somatic cells such as embryonic fibroblasts, amniotic cells, mammary cells, and Sertoli cells [38,39]. These reports showed that epigenetic disorders in bovine fibroblasts prevented complete reprogramming. Compared with bovine fibroblasts, bovine amniotic membrane-derived cells and mammary epithelial cells were more easily reprogrammed [66,67]. Bovine neural stem cells and testicular germ cells were reprogrammed into biPSC-like cells with *OCT4* alone [44,68,69].

However, despite numerous efforts, attempts to maintain biPSCs by promoting self-renewal have not been entirely successful. The currently obtained biPSC-like cells still do not meet all the criteria of PSCs. To some extent, these cells might present unstable expression of reprogramming factors, low reprogramming efficiency, the inability to maintain pluripotency after prolonged cultivation, and limited developmental potential in vitro and in vivo (inability to generate chimeras). Nevertheless, these works have laid a solid foundation for future studies of biPSCs, stem cell signaling pathways, and the generation of transgenic cattle. These findings represent advances in biPSC technology and broaden our understanding of bovine pluripotency molecules.

## 5. Establishment of bEPSCs

Chimeras can be used to test the developmental potential of donor cell lineages and are therefore considered the gold standard for assessing stem cell pluripotency [70]. Both bESC-like cells and biPSC-like cells have limited developmental potential to contribute to extraembryonic tissues, especially the placenta, and therefore they do not meet the gold standard. Unlike ESCs, EPSCs are derived from four- and eight-cell embryos. Zhao et al. successfully established wild-type and SCNT bEPSCs from preimplantation embryos in 2021 [71], based on the successive establishment of mouse, human, and porcine EPSCs. The bEPSCs express high levels of pluripotency genes and can be stably propagated in a feeder-free culture environment. Their pluripotent genes are expressed stably during long-term culture and can be edited effectively and precisely. Additionally, bEPSCs share a rich transcriptomic profile with early preimplantation embryos. They are capable of differentiation into cells of all three germ layers in vitro and into embryonic (fetal) and extraembryonic cell lineages in chimeras. Importantly, genetically modified bEPSCs can be used as donors for SCNT to significantly advance biotechnology and agriculture.

## 6. Challenges

The establishment of primed bESCs, although reported, provides limited information on the species–specific mechanisms that maintain the pluripotent state. There are still no reports on the in-depth mechanism of bovine pluripotency to help us easily establish bESCs. bESCs are still constrained by the difficulties of isolation, culture, and lineage establishment [26]. In studies of livestock ESCs, research on pig ESCs has recently made remarkable progress. A more in-depth study of pluripotency regulation in early bovine embryo that refers to the lineage establishment method of pig ESCs and combines it with the characteristics of bovine early embryonic development will help to stabilize the establishment of bESCs.

The generation of pluripotent cells by reprogramming has overcome limitations such as gender, age, and reproductive status, which is attractive for specific applications such as genetic preservation. However, methods known to increase reprogramming efficiency in mouse and human cells have not helped generate biPSCs. Thus, the biPSC-like cells reported are only partially reprogrammed cells, and establishing biPSCs remains challenging [72]. Viral integration during reprogramming can lead to karyotypic abnormalities and genomic alterations that reduce the efficiency of PSCs. Therefore, the derivation of iPSCs must be improved for reproducibility and safety [73]. Furthermore, it has not been investigated so far how transcription factors bind and function in biPSCs. It is conceivable that transcription factors may have different binding patterns in cells due to their different origins. They may play only a transient role during reprogramming, a role that cannot be analyzed by the final pluripotent state. Thus, the intermediate stages of the process need to be studied to understand the specific contribution of each factor to the reprogramming process. Together, the molecular mechanisms involved in the reprogramming process could be elucidated by analyzing the intermediate stages and the final stages. Reprogrammed somatic cells are mostly unable to silence exogenous factors; therefore, the derived biPSC-like cells usually carry them. These biPSC-like cells self-renew and maintain homeostasis mainly by these exogenous factors and also partially by endogenous factors [74]. This might be the reason why the authentic biPSC establishment failed. Therefore, it is urgent to explore how to completely activate endogenous pluripotency genes and silence exogenous factors simultaneously [75]. Currently, the culture system for biPSC-like cells cannot maintain the expression of endogenous pluripotent transcription factors. Pluripotency is one of the characteristics of iPSCs. The criteria for the identification of stem cell pluripotency include at least seven aspects: unlimited self-renewal, in vitro differentiation, teratoma assay, chimeric assay, germline assay, tetraploid compensation, and single-cell chimerism. At present, the actual pluripotency status (naïve, formative, and primed) of those biPSC-like cells still needs to be characterized. Thus, the differentiation research of biPSC-like cells is limited. 

## 7. Application Outlook

bPSCs may contribute to cattle breeding and reproduction [76]. Traditional livestock breeding takes a long time, while stem cell breeding might bypass this stage by obtaining bPSCs in vitro. bPSCs could be induced to differentiate into germ cells, and in turn, these germ cells could be used to produce fertilized eggs combined with in vitro fertilization and monosperm microinjection. Theoretically, this strategy can effectively shorten the intergenerational interval, improve the breeding efficiency, accelerate the breeding process, and facilitate the reproduction of good breeds to obtain genetically advantageous cattle with enhanced resistance to diseases. The bPSCs may also be used to produce cell-cultured meat, eliminating ecological threats to the farming industry and public health risks such as the spread of zoonoses. 

The in vitro differentiation of bPSCs can be used to model in vivo differentiation. The ability of bPSCs to recapitulate the developmental process in vitro allows their application in developmental biology research. By developing a three-dimensional culture system, like other stem cells bPSCs can give rise to structures that resemble whole organs (organoids), which shows great promise in modeling human development and diseases in biomedical research and regenerative medicine [77]. bPSCs might also be used as a tool for evolutionary analysis. The acquisition of multispecies iPSCs will help us to analyze interspecies cells in vitro, compare interspecies differences, and elucidate the process by which organisms acquire diversity and complexity.

The gradual accumulation of genetic mutations is the basis of cancer development. The similarity between somatic cell reprogramming and cancer initiation implies that there may be similar molecular mechanisms in these two processes. In addition, the mechanism of epigenetic instability triggered by the forced expression of reprogramming factors might also be used for research on cancer development. The advances in the two fields will contribute to a deeper understanding of cancer development.

biPSCs might also be valuable in disease modeling. Disease modeling refers to establishing experimental objects or related materials with human disease manifestations in medical research to promote our understanding of the mechanism by which diseases occur. Due to technical and financial constraints, most of the disease research is conducted using animal models. The employment of biPSCs will significantly reduce the use of experimental animals and protect animal welfare. For example, cows have a similar reproductive cycle to humans and might be able to model human ovarian and uterine diseases. It is also possible that biPSCs can be used for drug screening and gene editing. 

Although the research of bPSCs has broadened the application prospects in the above-mentioned fields, so far they are still uncertain. The key step in these possible applications is to establish the real bPSCs (bESCs/biPSCs/bEPSCs). Breaking through this bottleneck, bPSCs will bring more possibilities in the future.

## Figures and Tables

**Figure 1 ijms-25-02120-f001:**
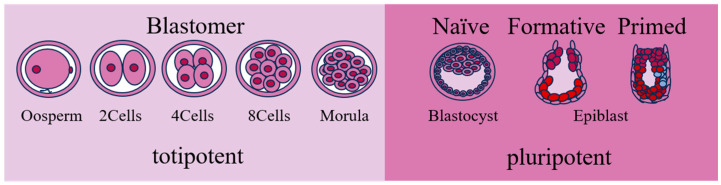
Early embryonic development. This indicates the characteristics of early embryonic development. Embryonic cells with totipotent or pluripotent potentials are accordingly categorized into different pluripotent states.

**Figure 2 ijms-25-02120-f002:**
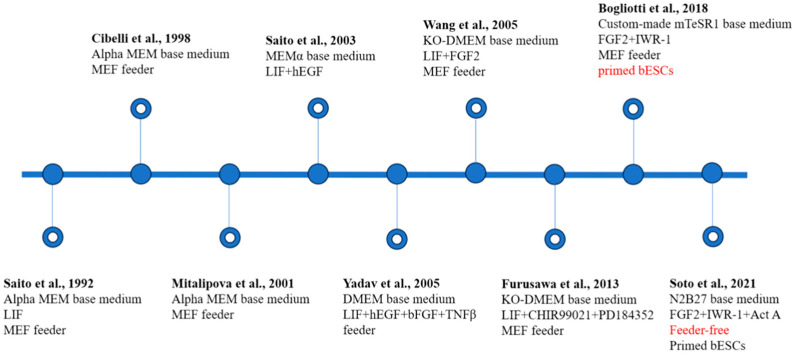
Establishment and characteristics of bovine embryonic stem cells (bESCs). The main components of the medium used and the pluripotent state of the cells are noted [17,21,25,26,27,28,29,30,31]. MEF, inactivated mouse embryonic fibroblast; KO-DMEM, knockout DMEM; Act-A, Activin A. Red text represents important breakthroughs.

## Data Availability

The data that support the findings of this study are available from the corresponding author upon reasonable request.

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
