# Peer review of "Bovine Pluripotent Stem Cells: Current Status and Prospects"

_ijms, 2024, doi:10.3390/ijms25042120_

Round 1
Reviewer 1 Report
Comments and Suggestions for Authors
The manuscript “Bovine pluripotent stem cells: current status and prospects” addresses very interesting field of research early embryonic development of different species. Although there are a lot of first glance similarities between mouse and rat development, rat’s pluripotent stem cells were generated only recently by reprogramming techniques. Thus generation of bovine pluripotent stem cells is still in progress. Overall, the manuscript is rather informative and could be interesting for the Readers, however I could not recommend in for publication in its present form. Some of the Authors statements should be explained or corrected.
General remarks. It is clear from the manuscript that bPSC lines that fit all pluripotency criteria are not currently established. Thus it would be better to refer to them as bPSC-like cell lines. In conclusion section Authors speculate that bPSC could be differentiated into germ cells however only a few controversial reports about mouse germ cells generation were published. Indeed, bPSCs could be probably differentiated, however it is unclear how they can model human disease. Should be thoroughly explained. The same concerns cancer modeling. I would recommend to rewrite the discussion section with regard on difficulties of bPSCs establishment.
1. Lines 26-27, 2nd sentence. ESCs are pluripotent stem cells existing only in vitro and were first established from mouse ICM in 1981.
2. Line 27 3rd sentence. Strictly speaking ESCs (cells existing only in vitro) are not able to develop different organ systems with functions. ESCs are able to develop only organoids- some organ-like structures in vitro.
3. Line 30. Cells could not be hypomethylated.
4. Line 31. “Unlike ectoderm pluripotent cells…” Why unlike ectoderm? Mesoderm will be different?
5. Line 36. “artificially transformed pluripotent stem cells”- what is it?
6. Line 43-44. “PSCs.. cannot develop into extraembryonic endoderm and trophoectoderm”. Human PSCs can. Authors have to consider it.
7. Line 63. “demethylation of iPSC…” . Cells could not be methylated or demethylated.
8. Line 94-95. ICM transition from naïve to primed state. ICM does not have these states and cited ref 12 does not describe it. Naïve and primed states are the characteristic of in vitro cultivated ESCs.
9. Line 107. Term transformation is usually used to indicate genetic alteration(s) leading to a new genotype (tumor or transgenic)
10. Line 126. “Mouse ectodermal stem cells (mEpiSCs)……” Incorrect, mouse epiblast stem cells.
11. Line 127. “mEpiSCs…have similar characteristics to hESCs”, but they are not equivalent to hESC because they are mouse.
12. Line 230-231. Sentence duplication
13. Line 236. “cell silencing”. What is it?
14. Line 260. “self-renewal and renewal” What is the difference?
15. Line 370-371. Sentence duplication.
Comments on the Quality of English LanguageMinor English editing
Reviewer 2 Report
Comments and Suggestions for Authors
The article reviews the status, challenges, and prospects of bovine pluripotent stem cells (bPSC). Also, the authors anticipate some potential applications for the bPSC. The topic might be potentially interesting; however, the paper does not review the bPSC. Instead of that, it looks like a short introduction with broad information and also describes many collateral topics and concepts without going deeper into the main topic, the bPSC. The Section 6 Challenge is an example of that. There are two paragraphs: first, there is no clear topic; second, it is about iPSCs and biPSCs, however, if we read the references and the text, it is just iPSCs. Therefore, I can conclude that the review has a very low quality and I recommend its rejection.
Comments on the Quality of English Language
The reviewers also want to highlight the difficulties found when reading the review. Some sentences are so long that is difficult to understand the subject and so on.
Round 2
Reviewer 1 Report
Comments and Suggestions for Authors
Authors considered all remarks and corrected the manuscript.
Reviewer 2 Report
Comments and Suggestions for Authors
The article reviews the status, challenges, and prospects of bovine pluripotent stem cells (bPSC), which is potentially interesting. It is not clear what the future or potential applications of the bPSC. However, the paper was clearly improved in terms of language and text structure. I still believe that this work does not review the bPSC in depth but I accept the current article.
Comments on the Quality of English LanguageThe language has been improved by the authors.